# Hierarchical Graph-to-Graph Translation for Molecules

## Abstract

The problem of accelerating drug discovery relies heavily on automatic tools to optimize precursor molecules to afford them with better biochemical properties. Our work in this paper substantially extends prior state-of-the-art on graph-to-graph translation methods for molecular optimization. In particular, we realize coherent multi-resolution representations by interweaving the encoding of substructure components with the atom-level encoding of the original molecular graph. Moreover, our graph decoder is fully autoregressive, and interleaves each step of adding a new substructure with the process of resolving its attachment to the emerging molecule. We evaluate our model on multiple molecular optimization tasks and show that our model significantly outperforms previous state-of-the-art baselines.

## 1 Introduction

Molecular optimization seeks to modify compounds in order to improve their biochemical properties. This task can be formulated as a graph-to-graph translation problem analogous to machine translation. Given a corpus of molecular pairs $\{(X, Y)\}$, where $Y$ is a paraphrase of $X$ with better chemical properties, the model is trained to translate an input molecular graph into its better form. The task is difficult since the space of potential candidates is vast, and molecular properties can be complex functions of structural features. Moreover, graph generation is computationally challenging due to complex dependencies involved in the joint distribution over nodes and edges. Similar to machine translation, success in this task is predicated on the inductive biases built into the encoder-decoder architecture, in particular the process of generating molecular graphs.

Prior work (Jin et al., 2019) proposed a junction tree encoder-decoder that utilized valid chemical substructures (e.g., aromatic rings) as building blocks to generate graphs. Each molecule was represented as a junction tree over chemical substructures in addition to the original atom-level graph. While successful, the approach remains limited in several ways. The tree and graph encoding were carried out separately, and decoding proceeded in strictly successive steps: first generating the junction tree for the new molecule, and then attaching its substructures together. This means the predicted attachments do not impact the subsequent substructure choices (see Figure 1a). Moreover, the attachment prediction process is non-autoregressive, thus it can predict inconsistent substructure attachments across different nodes in the junction tree (see Figure 1b).

We propose a multi-resolution, hierarchically coupled encoder-decoder for graph generation. Our auto-regressive decoder interleaves the prediction of substructure components with their attachments to the molecule being generated. In particular, a target graph is unraveled as a sequence of triplet predictions (where to expand the graph, new substructure type, its attachment). This enables us to model strong dependencies between successive attachments and substructure choices. The encoder is designed to represent molecules at different resolutions in order to match the proposed decoding process. Specifically, the encoding of each molecule proceeds across three levels, with each layer capturing essential information for its corresponding decoding step. The graph convolution of atoms at the lowest level supports the prediction of attachments and the convolution over substructures at the highest level supports the prediction of successive substructures. Compared to prior work, our decoding process is much more efficient because it decomposes each generation step into a hierarchy of smaller steps in order to avoid combinatorial explosion. We also extend the method to handle conditional translation where desired criteria are fed as input to the translation process. This enables our method to handle different combinations of criteria at test time.

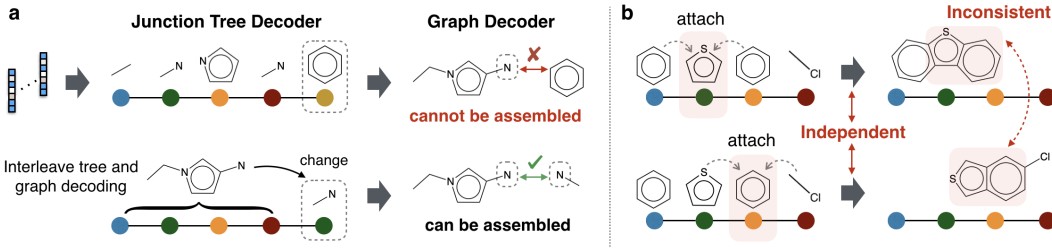

Figure 1: Key limitations of Jin et al. (2019)'s approach: a) Since their tree and graph decoders are isolated, the model can generate invalid junction trees which cannot be assembled into any molecule. This problem can be solved when we interleave the tree and graph decoding steps, allowing the predicted attachments to guide the substructure prediction; b) Their non-autoregressive graph decoder often predicts inconsistent local substructure attachments during training. To this end, we propose an autoregressive decoder that interleaves the prediction of substructures with their attachments.

We evaluate our new model on multiple molecular optimization tasks. Our baselines include previous state-of-the-art graph generation methods (You et al., 2018a; Liu et al., 2018; Jin et al., 2019) and an atom-based translation model we implemented for a more comprehensive comparison. Our model significantly outperforms these methods in discovering molecules with desired properties, yielding 3.3% and 8.1% improvement on QED and DRD2 optimization tasks. During decoding, our model runs 6.3 times faster than previous substructure-based generation methods. We further conduct ablation studies to validate the advantage of our hierarchical decoding and multi-resolution encoding. Finally, we show that conditional translation can succeed (generalize) even when trained on molecular pairs with only 1.6% of them having desired target property combination.

## 2  RELATED WORK

**Molecular Graph Generation**  Previous work have adopted various approaches for generating molecular graphs. Methods (Gómez-Bombarelli et al., 2018; Segler et al., 2017; Kusner et al., 2017; Dai et al., 2018; Guimaraes et al., 2017; Olivecrona et al., 2017; Popova et al., 2018; Kang & Cho, 2018) generate molecules based on their SMILES strings (Weininger, 1988). Simonovsky & Komodakis (2018); De Cao & Kipf (2018); Ma et al. (2018) developed generative models which output the adjacency matrices and node labels of the graphs at once. You et al. (2018b); Li et al. (2018); Samanta et al. (2018); Liu et al. (2018) proposed generative models decoding molecules sequentially node by node. You et al. (2018a); Zhou et al. (2018) adopted similar node-by-node approaches in the context of reinforcement learning. Kajino (2018) developed a hypergraph grammar based method for molecule generation.

Our work is most closely related to Jin et al. (2018; 2019) that generate molecules based on substructures. They adopted a two-stage procedure for realizing graphs. The first step generates a junction tree with substructures as nodes, capturing their coarse relative arrangements. The second step resolves the full graph by specifying how the substructures should be attached to each other. Their major drawbacks are 1) The second step introduced local independence assumptions and therefore the decoder is not autoregressive. 2) These two steps are applied stage-wise during decoding – first realizing the junction tree and then reconciling attachments without feedback. In contrast, our method jointly predicts the substructures and their attachments with an autoregressive decoder.

**Graph Encoders**  Graph neural networks have been extensively studied for graph encoding (Scarselli et al., 2009; Bruna et al., 2013; Li et al., 2015; Niepert et al., 2016; Kipf & Welling, 2017; Hamilton et al., 2017; Lei et al., 2017; Velickovic et al., 2017; Xu et al., 2018). Our method is related to graph encoders for molecules (Duvenaud et al., 2015; Kearnes et al., 2016; Dai et al., 2016; Gilmer et al., 2017; Schütt et al., 2017). Different to these approaches, our method represents molecules as hierarchical graphs spanning from atom-level graphs to substructure-level trees.

Our work is most closely related to (Defferrard et al., 2016; Ying et al., 2018; Gao & Ji, 2019) that learn to represent graphs in a hierarchical manner. In particular, Defferrard et al. (2016) utilized graph coarsening algorithms to construct multiple layers of graph hierarchy and Ying et al. (2018);

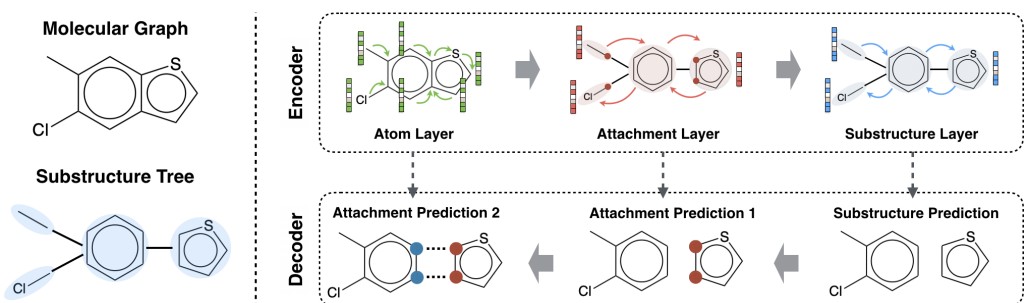

Figure 2: Overview of our approach. Each substructure $\mathcal{S}_i$ is a subgraph of a molecule (e.g., rings). In each step, our decoder adds a new substructure and predicts its attachment to current graph. Our encoder represents each molecule across three levels (atom layer, attachment layer and substructure layer), with each layer capturing relevant information for the corresponding decoding step.

Gao & Ji (2019) proposed to learn the graph hierarchy jointly with the encoding process. Despite some differences, all of these methods seek to represent graphs as a single vector for regression or classification tasks. In contrast, our focus is graph generation and a molecule is encoded into multiple sets of vectors, each representing the input at different resolutions. Those vectors are dynamically aggregated by decoder attention modules in each graph generation step.

## 3   HIERARCHICAL GENERATION OF MOLECULAR GRAPHS

The graph translation task seeks to learn a function $\mathcal{F}$ that maps a molecule $X$ into another molecule $\mathcal{G}$ with better chemical properties. $\mathcal{F}$ is parameterized as an encoder-decoder with neural attention. Both our encoder and decoder are illustrated in Figure 2. In each generation step, our decoder adds a new substructure (*substructure prediction*) and decides how it should be attached to the current graph. The attachment prediction proceeds in two steps: predicting attaching points in the new substructure and their corresponding attaching points in the current graph (*attachment prediction 1-2*).

To support the above hierarchical generation, we need to design a matching encoder representing molecules at multiple resolutions in order to provide necessary information for each decoding step. Therefore, we propose to represent a molecule $X$ by a hierarchical graph $\mathcal{H}_X$ with three components: 1) *substructure layer* representing how substructures are coarsely connected; 2) *attachment layer* showing the attachment configuration of each substructure; 3) *atom layer* showing how atoms are connected in the graph. Our model encodes nodes in $\mathcal{H}_X$ into substructure vectors $c_X^S$, attachment vectors $c_X^A$ and atom vectors $c_X^{\mathcal{G}}$, which are fed to the decoder for corresponding prediction steps. As our encoder is tailored for the decoder, we first describe our decoder to clarify relevant concepts.

### 3.1   HIERARCHICAL GRAPH DECODER

**Notations** We denote the sigmoid function as $\sigma(\cdot)$. $\mathrm{MLP}(\boldsymbol{a}, \boldsymbol{b})$ represents a multi-layer neural network whose input is the concatenation of $\boldsymbol{a}$ and $\boldsymbol{b}$. $\mathrm{attention}_\theta(\boldsymbol{h}_*, \boldsymbol{c}_X)$ stands for a bilinear attention over vectors $\boldsymbol{c}_X$ with query vector $\boldsymbol{h}_*$.

**Substructures** We define a substructure $\mathcal{S}_i = (\mathcal{V}_i, \mathcal{E}_i)$ as subgraph of molecule $\mathcal{G}$ induced by atoms in $\mathcal{V}_i$ and bonds in $\mathcal{E}_i$. Given a molecule, we extract its substructures $\mathcal{S}_1, \cdots, \mathcal{S}_n$ such that their union covers the entire molecular graph: $\mathcal{V} = \bigcup_i \mathcal{V}_i$ and $\mathcal{E} = \bigcup_i \mathcal{E}_i$. In this paper, we consider two types of substructures: rings and bonds. We denote the vocabulary of substructures as $\mathcal{S}$, which is constructed from the training set. In our experiments, $|\mathcal{S}| < 500$ and it has over 99.5% coverage on test sets.

**Substructure Tree** To characterize how substructures are connected in the molecule $\mathcal{G}$, we construct its corresponding substructure tree $\mathcal{T}$, whose nodes are substructures $\mathcal{S}_1, \cdots, \mathcal{S}_n$. Specifically, we construct the tree by first drawing edges between $\mathcal{S}_i$ and $\mathcal{S}_j$ if they share common atoms, and then applying tree decomposition over $\mathcal{T}$ to ensure it is tree-structured.

**Generation** Our graph decoder generates a molecule $\mathcal{G}$ by incrementally expanding its substructure tree in its depth-first order. Suppose the model is currently visiting substructure node $\mathcal{S}_k$. It makes the following predictions conditioned on encoding of input $X$ (see Figure 3):

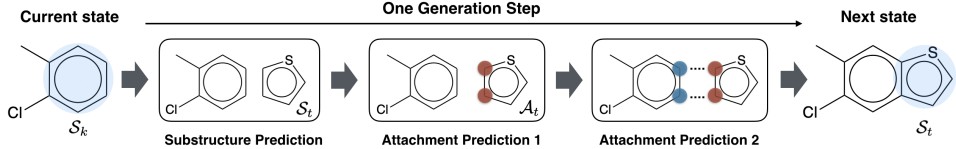

Figure 3: Illustration of hierarchical graph decoding. Suppose the decoder is visiting the substructure $\mathcal{S}_k$. 1) It decides to add a new substructure (*topological prediction*). 2) It predicts that new substructure $\mathcal{S}_t$ should be a ring (*substructure prediction*) 3) It predicts how this new ring should be attached to the graph (*attachment prediction*). Finally, the decoder moves to $\mathcal{S}_t$ and repeats the process.

1. **Topological Prediction**: It first predicts whether there will be a new substructure attached to $\mathcal{S}_k$. If not, the model backtracks to its parent node $\mathcal{S}_{d_k}$ in the tree. Let $\boldsymbol{h}_{\mathcal{S}_k}$ be the hidden representation of $\mathcal{S}_k$ learned by decoder (which will be elaborated in §3.2). This probability is predicted by a MLP with attention over substructure vectors $\boldsymbol{c}_X^{\mathcal{S}}$ of $X$:

$$\boldsymbol{p}_k = \sigma(\mathrm{MLP}(\boldsymbol{h}_{\mathcal{S}_k}, \boldsymbol{\alpha}_k^d)) \qquad \boldsymbol{\alpha}_k^d = \mathrm{attention}_d\left(\boldsymbol{h}_{\mathcal{S}_k}, \boldsymbol{c}_X^{\mathcal{S}}\right) \tag{1}$$

2. **Substructure Prediction**: If $\boldsymbol{p}_k > 0.5$, the model decides to create a new substructure $\mathcal{S}_t$ from $\mathcal{S}_k$ and sets its parent $d_t = k$. It then predicts the substructure type of $\mathcal{S}_t$ using another MLP that outputs a distribution over the vocabulary $\mathcal{S}$:

$$\boldsymbol{p}_{\mathcal{S}_t} = \mathrm{softmax}(\mathrm{MLP}(\boldsymbol{h}_{\mathcal{S}_k}, \boldsymbol{\alpha}_k^s)) \qquad \boldsymbol{\alpha}_k^s = \mathrm{attention}_s\left(\boldsymbol{h}_{\mathcal{S}_k}, \boldsymbol{c}_X^{\mathcal{S}}\right) \tag{2}$$

3. **Attachment Prediction**: Now the model needs to decide how $\mathcal{S}_t$ should be attached to $\mathcal{S}_k$. The attachment between $\mathcal{S}_t$ and $\mathcal{S}_k$ is defined as atom pairs $\mathcal{M}_t = \{(u_j, v_j)|u_j \in \mathcal{S}_t, v_j \in \mathcal{S}_k\}$ where atom $u_j$ and $v_j$ are attached together. We predict those atom pairs in two steps:

   1) We first predict the atoms $\{v_j\} \subset \mathcal{S}_t$ that will be attached to $\mathcal{S}_k$. Since the graph $\mathcal{S}_t$ is always fixed and the number of attaching atoms between two substructures is usually small, we can enumerate all possible configurations $\{v_j\}$ to form a vocabulary $\mathcal{A}(\mathcal{S}_t)$ for each substructure $\mathcal{S}_t$. This allows us to formulate the prediction of $\{v_j\}$ as a classification task – predicting the correct configuration $\mathcal{A}_t = (\mathcal{S}_t, \{v_j\})$ from the vocabulary $\mathcal{A}(\mathcal{S}_t)$:

   $$\boldsymbol{p}_{\mathcal{A}_t} = \mathrm{softmax}(\mathrm{MLP}(\boldsymbol{h}_{\mathcal{S}_k}, \boldsymbol{\alpha}_k^a)) \qquad \boldsymbol{\alpha}_k^a = \mathrm{attention}_a\left(\boldsymbol{h}_{\mathcal{S}_k}, \boldsymbol{c}_X^{\mathcal{A}}\right) \tag{3}$$

   2) Given the predicted attaching points $\{v_j\}$, we need to find the corresponding atoms $\{u_j\}$ in the substructure $\mathcal{S}_k$. As the attaching points are always consecutive, there exist at most $2|\mathcal{S}_k|$ different attachments $M = \{(u_j, v_j)\}$. The probability of a candidate attachment $M$ is computed based on the atom representations $\boldsymbol{h}_{u_j}$ and $\boldsymbol{h}_{v_j}$ learned by the decoder:

   $$\boldsymbol{p}_M = \mathrm{softmax}\left(\boldsymbol{h}_M \cdot \mathrm{attention}_m(\boldsymbol{h}_M, \boldsymbol{c}_X^{\mathcal{G}})\right) \quad \boldsymbol{h}_M = \sum_j \mathrm{MLP}(\boldsymbol{h}_{u_j}, \boldsymbol{h}_{v_j}) \tag{4}$$

The above three predictions together give an autoregressive factorization of the distribution over the next substructure and its attachment. Each of the three decoding steps depends on the outcome of previous step, and predicted attachments will in turn affect the prediction of subsequent substructures. During training, we apply teacher forcing to the above generation process, where the generation order is determined by a depth-first traversal over the ground truth substructure tree. The attachment enumeration is tractable because most of the substructures are small. In our experiments, the average size of attachment vocabulary $|\mathcal{A}(\mathcal{S}_t)| < 5$ and the number of candidate attachments is less than 20.

### 3.2 HIERARCHICAL GRAPH ENCODER

Our encoder represents a molecule $X$ by a hierarchical graph $\mathcal{H}_X$ in order to support the above decoding process. The hierarchical graph has three components (see Figure 4):

1. **Atom layer**: The atom layer is the molecular graph of $X$ representing how its atoms are connected. Each atom node $v$ is associated with a label $a_v$ indicating its atom type and charge. Each edge $(u, v)$ in the atom layer is labeled with $b_{uv}$ indicating its bond type.
2. **Attachment layer**: This layer is derived from the substructure tree of molecule $X$. Each node $\mathcal{A}_i$ in this layer represents a particular attachment configuration of substructure $\mathcal{S}_i$ in the vocabulary $\mathcal{A}(\mathcal{S}_i)$. Specifically, $\mathcal{A}_i = (\mathcal{S}_i, \{v_j\})$ where $\{v_j\}$ are the attaching atoms between $\mathcal{S}_i$ and its parent $\mathcal{S}_{d_i}$ in the tree. This layer provides necessary information for the attachment prediction (step 1). Figure 4 illustrates how $\mathcal{A}_i$ and the vocabulary $\mathcal{A}(\mathcal{S}_i)$ look like.

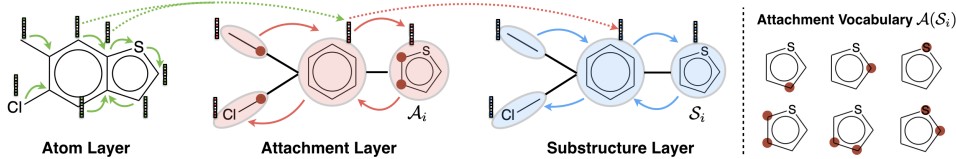

Figure 4: **Left**: Hierarchical graph encoder. Solid arrows illustrate message passing in each layer. Dashed arrows connect each atom to the substructures it belongs. In the attachment layer, each node $\mathcal{A}_i$ is a particular attachment configuration of substructure $\mathcal{S}_i$. **Right**: Attachment vocabulary for a ring. The attaching points in each configuration (highlighted in red) must be consecutive.

3. **Substructure layer**: This layer is the same as the substructure tree. This layer provides essential information for the substructure prediction in the decoding process.

We further introduce edges that connect the atoms and substructures between different layers in order to propagate information in between. In particular, we draw a directed edge from atom $v$ in the atom layer to node $\mathcal{A}_i$ in the attachment layer if $v \in \mathcal{S}_i$. We also draw edges from node $\mathcal{A}_i$ to node $\mathcal{S}_i$ in the substructure layer. This gives us the hierarchical graph $\mathcal{H}_X$ for molecule $X$, which will be encoded by a hierarchical message passing network (MPN) (see Figure 4). The encoder contains three MPNs that encode each of the three layer. We use the MPN architecture from Jin et al. (2019).[1] For simplicity, we denote the MPN encoding process as $\mathrm{MPN}_\psi(\cdot)$ with parameter $\psi$.

**Atom Layer MPN** We first encode the atom layer of $\mathcal{H}_X$ (denoted as $\mathcal{H}_X^g$). The inputs to this MPN are the embedding vectors $\{e(a_u)\}, \{e(b_{uv})\}$ of all the atoms and bonds in $X$. During encoding, the network propagates the message vectors between different atoms for $T$ iterations and then outputs the atom representation $h_v$ for each atom $v$:

$$c_X^{\mathcal{G}} = \{h_v\} = \mathrm{MPN}_{\psi_1}\left(\mathcal{H}_X^g, \{e(a_u)\}, \{e(b_{uv})\}\right) \tag{5}$$

**Attachment Layer MPN** The input feature of each node $\mathcal{A}_i$ in the attachment layer $\mathcal{H}_X^a$ is an concatenation of the embedding $e(\mathcal{A}_i)$ and the sum of its atom vectors $\{h_v \mid v \in \mathcal{S}_i\}$:

$$f_{\mathcal{A}_i} = \mathrm{MLP}\left(e(\mathcal{A}_i), \sum\nolimits_{v \in \mathcal{S}_i} h_v\right) \tag{6}$$

The input feature for each edge $(\mathcal{A}_i, \mathcal{A}_j)$ in this layer is an embedding vector $e(d_{ij})$, where $d_{ij}$ describes the relative ordering between node $\mathcal{A}_i$ and $\mathcal{A}_j$ during decoding. Specifically, we set $d_{ij} = k$ if node $\mathcal{A}_i$ is the $k$-th child of node $\mathcal{A}_j$ and $d_{ij} = 0$ if $\mathcal{A}_i$ is the parent. We then run $T$ iterations of message passing over $\mathcal{H}_X^a$ to compute the substructure representations:

$$c_X^{\mathcal{A}} = \{h_{\mathcal{A}_i}\} = \mathrm{MPN}_{\psi_2}\left(\mathcal{H}_X^a, \{f_{\mathcal{A}_i}\}, \{e(d_{ij})\}\right) \tag{7}$$

**Substructure Layer MPN** Similarly, the input feature of node $\mathcal{S}_i$ in this layer is computed as the concatenation of embedding $e(\mathcal{S}_i)$ and the node vector $h_{\mathcal{A}_i}$ from the previous layer. Finally, we run message passing over the substructure layer $\mathcal{H}_X^s$ to obtain the substructure representations:

$$f_{\mathcal{S}_i} = \mathrm{MLP}\left(e(\mathcal{S}_i), h_{\mathcal{A}_i}\right) \qquad c_X^{\mathcal{S}} = \{h_{\mathcal{S}_i}\} = \mathrm{MPN}_{\psi_3}\left(\mathcal{H}_X^s, \{f_{\mathcal{S}_i}\}, \{e(d_{ij})\}\right) \tag{8}$$

In summary, the output of our hierarchical encoder is a set of vectors $c_X = c_X^{\mathcal{S}} \cup c_X^{\mathcal{A}} \cup c_X^{\mathcal{G}}$ that represent a molecule $X$ at multiple resolutions. These vectors are input to the decoder attention.

**Decoder MPN** During decoding, we use the same hierarchical MPN architecture to encode the hierarchical graph $\mathcal{H}_{\mathcal{G}}$ at each step $t$. This gives us the substructure vectors $h_{\mathcal{S}_k}$ and atom vectors $h_{v_j}$ in §3.1. All future nodes and edges are masked to ensure the prediction of current substructure and attachment only depends on previously generated outputs.

### 3.3 Training

Our training set contains molecular pairs $(X, Y)$ where each compound $X$ can be associated with multiple outputs $Y$ since there are many ways to modify $X$ to improve its properties. In order to

---

[1]We slightly modified their architecture by using LSTM instead of GRU for message propagation due to its better empirical performance. The details are shown in the appendix.

**Algorithm 1** Variational Translation (unconditional setting, without target criteria $\boldsymbol{g}$)

---
1: **for** $(X, Y)$ in the training set **do**
2:     Encode molecule $X, Y$ into vectors $\boldsymbol{c}_X, \boldsymbol{c}_Y$
3:     Compute $\boldsymbol{\mu}_{X,Y}, \boldsymbol{\sigma}_{X,Y}$ from $\boldsymbol{\delta}_{X,Y}$
4:     Sample latent code $\boldsymbol{z} \sim Q(\boldsymbol{z}|X,Y)$
5:     Generate molecule $Y$ given $\boldsymbol{c}_X$ and $\boldsymbol{z}$
6: **end for**

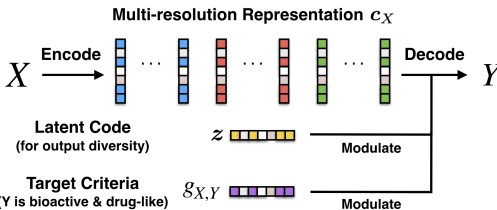

Figure 5: Conditional translation.

generate diverse outputs, we follow Jin et al. (2019) and extend our method to a variational translation model $\mathcal{F} : (X, \boldsymbol{z}) \rightarrow Y$ with an additional input $\boldsymbol{z}$. The latent vector $\boldsymbol{z}$ indicates the intended mode of translation which is sampled from a Gaussian prior $P(\boldsymbol{z})$ during testing.

We train our model using variational inference (Kingma & Welling, 2013). Given a training example $(X, Y)$, we sample $\boldsymbol{z}$ from the posterior $Q(\boldsymbol{z}|X,Y) = \mathcal{N}(\boldsymbol{\mu}_{X,Y}, \boldsymbol{\sigma}_{X,Y})$. To compute $Q(\boldsymbol{z}|X,Y)$, we first encode $X$ and $Y$ into their representations $\boldsymbol{c}_X$ and $\boldsymbol{c}_Y$ and then compute vector $\boldsymbol{\delta}_{X,Y}$ that summarizes the structural changes from molecule $X$ to $Y$ at both atom and substructure level:

$$\boldsymbol{\delta}_{X,Y}^{\mathcal{S}} = \sum \boldsymbol{c}_Y^{\mathcal{S}} - \sum \boldsymbol{c}_X^{\mathcal{S}} \qquad \boldsymbol{\delta}_{X,Y}^{\mathcal{G}} = \sum \boldsymbol{c}_Y^{\mathcal{G}} - \sum \boldsymbol{c}_X^{\mathcal{G}} \qquad (9)$$

Finally, we compute $[\boldsymbol{\mu}_{X,Y}, \boldsymbol{\sigma}_{X,Y}] = \mathrm{MLP}(\boldsymbol{\delta}_{X,Y}^{\mathcal{S}}, \boldsymbol{\delta}_{X,Y}^{\mathcal{G}})$ and sample $\boldsymbol{z}$ using reparameterization trick. The latent code $\boldsymbol{z}$ is passed to the decoder along with the input representation $\boldsymbol{c}_X$ to reconstruct output $Y$. The overall training objective follows a standard conditional VAE:

$$\mathcal{L}(X, Y) = -\mathbb{E}_{\boldsymbol{z} \sim Q}[\log P(Y|\boldsymbol{z}, X)] + \lambda_{\mathrm{KL}} \mathcal{D}_{\mathrm{KL}}[Q(\boldsymbol{z}|X,Y)||P(\boldsymbol{z})] \qquad (10)$$

**Conditional Translation** In the above formulation, the model does not know what properties are being optimized during translation. During testing, users cannot change the behavior of a trained model (i.e., what properties should be changed). This may become a limitation of our method in a multi-property optimization setting. Therefore, we extend our method to handle conditional translation where the desired criteria are also fed as input to the translation process. In particular, let $\boldsymbol{g}_{X,Y}$ be a translation criteria indicating what properties should be changed. During variational inference, we compute $\boldsymbol{\mu}_{X,Y}$ and $\boldsymbol{\sigma}_{X,Y}$ with an additional input $\boldsymbol{g}_{X,Y}$:

$$[\boldsymbol{\mu}_{X,Y}, \boldsymbol{\sigma}_{X,Y}] = \mathrm{MLP}(\boldsymbol{\delta}_{X,Y}^{\mathcal{S}}, \boldsymbol{\delta}_{X,Y}^{\mathcal{G}}, \boldsymbol{g}_{X,Y}) \qquad (11)$$

We then augment the latent code as $[\boldsymbol{z}, \boldsymbol{g}_{X,Y}]$ and pass it to the decoder. During testing, the user can specify their criteria in $\boldsymbol{g}_{X,Y}$ to control the outcome (e.g., $Y$ should be drug-like and bioactive).

## 4 EXPERIMENTS

We follow the experimental design by Jin et al. (2019) and evaluate our translation model on their single-property optimization tasks. As molecular optimization in the real-world often involves different property criteria, we further construct a novel conditional optimization task where the desired criteria is fed as input to the translation process. To prevent the model from ignoring input $X$ and translating it into arbitrary compound, we require the molecular similarity between $X$ and output $Y$ to be above certain threshold $\mathrm{sim}(X,Y) \geq \delta$ at test time. The molecular similarity is defined as the Tanimoto similarity over Morgan fingerprints (Rogers & Hahn, 2010) of two molecules.

**Single-property Optimization** This dataset consists of four different tasks. For each task, we train and evaluate our model on their provided training and test sets. For these tasks, our model is trained under an unconditional setting (without $\boldsymbol{g}_{X,Y}$ as input).

- **LogP Optimization**: The penalized logP score (Kusner et al., 2017) measures the solubility and synthetic accessibility of a compound. In this task, the model needs to translate input $X$ into output $Y$ such that $\mathrm{logP}(Y) > \mathrm{logP}(X)$. We experiment with two similarity thresholds $\delta = \{0.4, 0.6\}$.
- **QED Optimization**: The QED score (Bickerton et al., 2012) quantifies a compound's drug-likeness. In this task, the model is required to translate molecules with QED scores from the lower range $[0.7, 0.8]$ into the higher range $[0.9, 1.0]$. The similarity constraint is $\mathrm{sim}(X,Y) \geq 0.4$.

Table 1: Results on single-property translation tasks. "Div." stands for diversity. "Succ." stands for success rate. "Improve." stands for average property improvement.

| Method | logP (sim $\geq$ 0.6) | | logP (sim $\geq$ 0.4) | | QED | | DRD2 | |
|---|---|---|---|---|---|---|---|---|
| | Improve. | Div. | Improve. | Div. | Succ. | Div. | Succ. | Div. |
| JT-VAE | $0.28 \pm 0.79$ | - | $1.03 \pm 1.39$ | - | 8.8% | - | 3.4% | - |
| CG-VAE | $0.25 \pm 0.74$ | - | $0.61 \pm 1.09$ | - | 4.8% | - | 2.3% | - |
| GCPN | $0.79 \pm 0.63$ | - | $2.49 \pm 1.30$ | - | 9.4% | 0.216 | 4.4% | 0.152 |
| MMPA | $1.65 \pm 1.44$ | 0.329 | $3.29 \pm 1.12$ | 0.496 | 32.9% | 0.236 | 46.4% | **0.275** |
| Seq2Seq | $2.33 \pm 1.17$ | 0.331 | $3.37 \pm 1.75$ | 0.471 | 58.5% | 0.331 | 75.9% | 0.176 |
| JTNN | $2.33 \pm 1.24$ | 0.333 | $3.55 \pm 1.67$ | 0.480 | 59.9% | 0.373 | 77.8% | 0.156 |
| AtomG2G | $2.41 \pm 1.19$ | 0.379 | $\mathbf{3.98 \pm 1.54}$ | 0.563 | 73.6% | 0.421 | 75.8% | 0.128 |
| HierG2G | $\mathbf{2.49 \pm 1.09}$ | **0.381** | $\mathbf{3.98 \pm 1.46}$ | **0.564** | **76.9%** | **0.477** | **85.9%** | 0.192 |

- **DRD2 Optimization**: This task involves the optimization of a compound's biological activity against dopamine type 2 receptor (DRD2). The model needs to translate inactive compounds ($p < 0.05$) into active compounds ($p \geq 0.5$), where the bioactivity is assessed by a property prediction model from Olivecrona et al. (2017). The similarity constraint is $\text{sim}(X, Y) \geq 0.4$.

**Conditional Optimization** This new task requires the model to translate input $X$ into output $Y$ to satisfy different combination of constraints over its QED and DRD2 scores. We define a molecule $Y$ as drug-like if $\text{QED}(Y) \geq 0.9$ and as DRD2-active if its predicted bioactivity $\text{DRD2}(Y) \geq 0.5$. At test time, our model needs to handle the following two criteria over output molecule $Y$:

1. $Y$ is both drug-like and DRD2-active. Here both properties need to be improved after translation.
2. $Y$ is drug-like but DRD2-inactive. In this case, DRD2 is an off-target that may cause side effects. Therefore only the drug-likeness should be improved after translation.

As different users may be interested in different settings, we encode the desired criteria as vector $\boldsymbol{g}$ and train our model under the conditional translation setup in §3.3. Like single-property tasks, we impose a similarity constraint $\text{sim}(X, Y) \geq 0.4$ for both settings.

Our training set contains 120K molecular pairs and the test set has 780 compounds. For each pair $(X, Y)$, we set $\boldsymbol{g}_{X,Y} = (\mathbb{I}[Y \text{ is drug-like}], \mathbb{I}[Y \text{ is DRD2-active}])$. During testing, we translate each compound with $\boldsymbol{g} = [1, 1], [1, 0]$ for each setting. We note that the first criteria ($\boldsymbol{g} = [1, 1]$) is the most challenging because there are only 1.6% of the training pairs with target $Y$ being both drug-like and DRD2-active. To achieve good performance, the model must learn to transfer the knowledge from other pairs with $\boldsymbol{g}_{X,Y} = [1, 0], [0, 1]$) that partially satisfy the criteria.

**Evaluation Metrics** Our evaluation metrics include translation accuracy and diversity. Each test molecule $X_i$ is translated $K = 20$ times with different latent codes sampled from the prior distribution. On the logP optimization, we select compound $Y_i$ as the final translation of $X_i$ that gives the highest property improvement and satisfies $\text{sim}(X_i, Y_i) \geq \delta$. We then report the average property improvement $\frac{1}{\mathcal{D}} \sum_i \log\text{P}(Y_i) - \log\text{P}(X_i)$ over test set $\mathcal{D}$. For other tasks, we report the translation success rate. A compound is successfully translated if one of its $K$ translation candidates satisfies all the similarity and property constraints of the task. To measure the diversity, for each molecule we compute the average pairwise Tanimoto distance between all its successfully translated compounds. Here the Tanimoto distance is defined as $\text{dist}(X, Y) = 1 - \text{sim}(X, Y)$.

**Baselines** We compare our method (HierG2G) against the baselines including GCPN (You et al., 2018a), MMPA (Dalke et al., 2018) and translation based methods Seq2Seq and JTNN (Jin et al., 2019). Seq2Seq is a sequence-to-sequence model that generates molecules by their SMILES strings. JTNN is a graph-to-graph architecture that generates molecules structure by structure, but its decoder is not fully autoregressive. We also compare with CG-VAE (Liu et al., 2018), a generative model that decodes molecules atom by atom and optimizes properties in the latent space using gradient ascent.

To make a direct comparison possible between our method and atom-based generation, we further developed an atom-based translation model (AtomG2G) as baseline. It makes three predictions in each generation step. First, it predicts whether the decoding process has completed (no more new

Table 2: Results on conditional optimization tasks and ablation studies over architecture choices.

(a) Conditional optimization results: $\boldsymbol{g} = [1, *]$ means the output $Y$ needs to be drug-like and $\boldsymbol{g} = [*, 1]$ means it needs to be DRD2-active.

| Method | $\boldsymbol{g} = [1, 1]$ | | $\boldsymbol{g} = [1, 0]$ | |
|---|---|---|---|---|
| | Succ. | Div. | Succ. | Div. |
| Seq2Seq | 5.0% | 0.078 | 67.8% | 0.380 |
| JTNN | 11.1% | 0.064 | 71.4% | 0.405 |
| AtomG2G | 12.5% | 0.031 | 74.5% | 0.443 |
| HierG2G | **13.0%** | **0.094** | **78.5%** | **0.480** |

(b) Ablation study: the importance of hierarchical graph encoding, LSTM MPN architecture and structure-based decoding.

| Method | QED | DRD2 |
|---|---|---|
| HierG2G | **76.9%** | **85.9%** |
| · atom-based decoder | 76.1% | 75.0% |
| · two-layer encoder | 75.8% | 83.5% |
| · one-layer encoder | 67.8% | 74.1% |
| · GRU MPN | 72.6% | 83.7% |

atoms). If not, it creates a new atom $a_t$ and predicts its atom type. Lastly, it predicts the bond type between $a_t$ and other atoms autoregressively to fully capture edge dependencies (You et al., 2018b). The encoder of AtomG2G encodes only the atom-layer graph and the decoder attention only sees the atom vectors $c_X^{\mathcal{G}}$. All translation models are trained under the same variational objective (§3.3).

**Single-property Optimization Results**  As shown in Table 1, our model achieves the new state-of-the-art on the four translation tasks. In particular, our model significantly outperforms JTNN in both translation accuracy (e.g., 76.9% versus 59.9% on the QED task) and output diversity (e.g., 0.564 versus 0.480 on the logP task). While both methods generate molecules by structures, our decoder is autoregressive which can learn more expressive mappings. More importantly, our model runs 6.3 times faster than JTNN during decoding. Our model also outperforms AtomG2G on three datasets, with over 10% improvement on the DRD2 task. This shows the advantage of our hierarchical model.

**Conditional Optimization Results**  For this task, we compare our method with other translation methods: Seq2Seq, JTNN and AtomG2G. All these models are trained under the conditional translation setup where we feed the desired criteria $\boldsymbol{g}_{X,Y}$ as input. As shown in Table 2a, our model outperforms other models in both translation accuracy and output diversity. Notably, all models achieved very low success rate on $\boldsymbol{c} = [1, 1]$ because it has the strongest constraints and only 1.6K of the training pairs satisfy this criteria. In fact, training our model on the 1.6K examples only gives 4.2% success rate as compared to 13.0% when trained with other pairs. This shows our conditional translation setup can transfer the knowledge from other pairs with $\boldsymbol{g}_{X,Y} = [1, 0], [0, 1]$.

**Ablation Study**  To understand the importance of different architecture choices, we report ablation studies over the QED and DRD2 tasks in Table 2b. We first replace our hierarchical decoder with atom-based decoder of AtomG2G to see how much the structure-based decoding benefits us. We keep the same hierarchical encoder but modified the input of the decoder attention to include both atom and substructure vectors. Using this setup, the model performance decreases by 0.8% and 10.9% on the two tasks. We suspect the DRD2 task benefits more from structure-based decoding because biological target binding often depends on the presence of specific functional groups.

Our second experiment reduces the number of hierarchies in our encoder and decoder MPN, while keeping the same hierarchical decoding process. When the top substructure layer is removed, the translation accuracy drops slightly by 0.8% and 2.4%. When we further remove the attachment layer, the performance degrades significantly on both datasets. This is because all the substructure information is lost and the model needs to infer what substructures are and how substructure layers are constructed for each molecule. Implementation details of those ablations are shown in the appendix. Lastly, we replaced our LSTM MPN with the original GRU MPN used in JTNN. While the translation performance decreased by 4% and 2.2%, our method still outperforms JTNN by a wide margin. Therefore we use the LSTM MPN architecture for both HierG2G and AtomG2G baseline.

## 5  CONCLUSION

In this paper, we developed a hierarchical graph-to-graph translation model that generates molecular graphs using chemical substructures as building blocks. In contrast to previous work, our model is fully autoregressive and learns coherent multi-resolution representations. The experimental results show that our method outperforms previous models under various settings.

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

## A ADDITIONAL FIGURES

$$g = [1,1]$$

(QED=0.841, DRD2=2e-4)      (QED=0.911, DRD2=0.964)

$$g = [1,0]$$

(QED=0.914, DRD2=6e-4)

Figure 6: Illustration of conditional translation. Our model generates different molecules when the translation criteria changes. When $\boldsymbol{g} = [1, 1]$, the model indeed generates a compound with high QED and DRD2 scores. When $\boldsymbol{g} = [1, 0]$, the model predicts another compound inactive to DRD2.

## B NETWORK ARCHITECTURE

**LSTM MPN Architecture** The LSTM MPN is a slight modification from the MPN architecture used in Jin et al. (2019). Let $N(v)$ be the neighbors of node $v$, $\boldsymbol{x}_v$ the node feature of $v$ and $\boldsymbol{x}_{uv}$ be the feature of edge $(u, v)$. During encoding, each edge $(u, v)$ is associated with two messages $\boldsymbol{\nu}_{uv}$ and $\boldsymbol{\nu}_{vu}$, representing the message from $u$ to $v$ and vice versa. The messages are updated by an LSTM cell with parameters $\psi = \{\boldsymbol{W}_\psi^z, \boldsymbol{W}_\psi^o, \boldsymbol{W}_\psi^r, \boldsymbol{W}_\psi\}$ defined as follows:

---

**Algorithm 2** LSTM Message Passing

**function** $\text{LSTM}_\psi \left( \boldsymbol{x}_u, \boldsymbol{x}_{uv}, \{\boldsymbol{\nu}_{wu}^{(t)}, \boldsymbol{c}_{wu}^{(t)}\}_{w \in N(u) \setminus v} \right)$

$$\boldsymbol{i}_{uv} = \sigma \left( \boldsymbol{W}_\psi^z \left[ \boldsymbol{x}_u, \boldsymbol{x}_{uv}, \sum_w \boldsymbol{\nu}_{wu}^{(t)} \right] + \boldsymbol{b}^z \right)$$

$$\boldsymbol{o}_{uv} = \sigma \left( \boldsymbol{W}_\psi^o \left[ \boldsymbol{x}_u, \boldsymbol{x}_{uv}, \sum_w \boldsymbol{\nu}_{wu}^{(t)} \right] + \boldsymbol{b}^o \right)$$

$$\boldsymbol{f}_{wu} = \sigma \left( \boldsymbol{W}_\psi^r \left[ \boldsymbol{x}_u, \boldsymbol{x}_{uv}, \boldsymbol{\nu}_{wu}^{(t)} \right] + \boldsymbol{b}^r \right)$$

$$\boldsymbol{c}_{uv}^{(t+1)} = \boldsymbol{i}_{uv} \odot \tanh \left( \boldsymbol{W}_\psi \left[ \boldsymbol{x}_u, \boldsymbol{x}_{uv}, \sum_w \boldsymbol{\nu}_{wu}^{(t)} \right] + \boldsymbol{b} \right) + \sum_w \boldsymbol{f}_{wu} \odot \boldsymbol{c}_{wu}^{(t)}$$

$$\boldsymbol{\nu}_{uv}^{(t+1)} = \boldsymbol{o}_{uv} \odot \tanh \left( \boldsymbol{c}_{uv}^{(t+1)} \right)$$

**Return** $\boldsymbol{\nu}_{uv}^{(t+1)}, \boldsymbol{c}_{uv}^{(t+1)}$

**end function**

---

The message passing network $\text{MPN}_\psi \left( \mathcal{H}, \{\boldsymbol{x}_u\}, \{\boldsymbol{x}_{uv}\} \right)$ over graph $\mathcal{H}$ is defined as:

---

**Algorithm 3** LSTM MPN with $T$ message passing iterations

**function** $\text{MPN}_\psi \left( \mathcal{H}, \{\boldsymbol{x}_v\}, \{\boldsymbol{x}_{uv}\} \right)$

  Initialize messages: $\boldsymbol{\nu}_{uv}^0 = \boldsymbol{0}, \boldsymbol{c}_{uv}^0 = \boldsymbol{0}$

  **for** $t = 0$ **to** $T - 1$ **do**

    Compute messages $\boldsymbol{\nu}_{uv}^{(t+1)}, \boldsymbol{c}_{uv}^{(t+1)} = \text{LSTM}_\psi \left( \boldsymbol{x}_u, \boldsymbol{x}_{uv}, \{\boldsymbol{\nu}_{wu}^{(t)}, \boldsymbol{c}_{wu}^{(t)}\}_{w \in N(u) \setminus v} \right)$ for all edges

    $(u, v) \in \mathcal{H}$ simultaneously.

  **end for**

  **Return** node representations $\boldsymbol{h}_v = \text{MLP} \left( \boldsymbol{x}_v, \sum_{u \in N(v)} \boldsymbol{\nu}_{uv}^{(T)} \right)$

**end function**

---

**Attention Layer** Our attention layer is a bilinear attention function with parameter $\theta = \{\boldsymbol{A}_\theta\}$:

$$\text{attention}_\theta(\boldsymbol{v}, \{\boldsymbol{h}_i\}) = \sum_i \beta_i \boldsymbol{h}_i \qquad \beta_i = \frac{\exp(\boldsymbol{v}^T \boldsymbol{A}_\theta \boldsymbol{h}_i)}{\sum_j \exp(\boldsymbol{v}^T \boldsymbol{A}_\theta \boldsymbol{h}_j)} \tag{12}$$

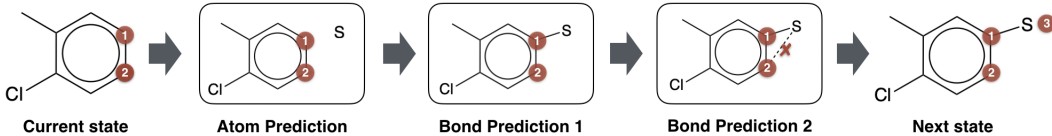

Figure 7: Illustration of AtomG2G decoding process. Atoms marked with red circles are frontier nodes in the queue $\mathcal{Q}$. In each step, the model picks the first node $v_t$ from $\mathcal{Q}$ and predict whether there will be new atoms attached to $v_t$. If so, it predicts the atom type of new node $u_t$ (atom prediction). Then the model predicts the bond type between $u_t$ and other nodes in $\mathcal{Q}$ sequentially for $|\mathcal{Q}|$ steps (bond prediction, $|\mathcal{Q}| = 2$). Finally, it adds the new atom to the queue $\mathcal{Q}$.

**AtomG2G Architecture** AtomG2G is an atom-based translation method that is directly comparable to HierG2G. Here molecules are represented solely as molecular graphs rather than a hierarchical graph with substructures. The encoder of AtomG2G is the same LSTM MPN over molecular graph. This gives us a set of atom vectors $c_X^{\mathcal{G}}$ representing molecule $X$ only at the atom level.

The decoder of AtomG2G is illustrated in Figure 7. Following You et al. (2018b); Liu et al. (2018), the model generates molecule $\mathcal{G}$ atom by atom following their breath-first order. During generation, it maintains a FIFO queue $\mathcal{Q}$ that contains the frontier nodes in the graph (i.e., nodes who still have neighbors to be generated). Let $v_t$ be the first node in $\mathcal{Q}$ and $\mathcal{G}_t$ be the current graph at step $t$. In each step, the model makes three predictions to expand the graph $\mathcal{G}_t$:

1. It predicts whether there will be new atoms attached to $v_t$. If not, the model discards $v$ and move on to the next node in $\mathcal{Q}$. The generation stops if $\mathcal{Q}$ is empty.
2. Otherwise, it creates a new atom $u_t$ and predicts its atom type.
3. Lastly, it predicts the bond type between $u_t$ and other frontier nodes in $\mathcal{Q}$ autoregressively to fully capture edge dependencies (You et al., 2018b). Since nodes are generated in breath-first order, there will be no edges between $u_t$ and nodes outside of $\mathcal{Q}$.

To make those predictions, we use the same LSTM MPN to encode the current graph $\mathcal{G}_t$. Let $h_{v_t}$ be the atom representation of $v_t$. We represent $\mathcal{G}_t$ as the sum of all its atom vectors $h_{\mathcal{G}_t} = \sum_{v \in \mathcal{G}_t} h_v$. In the first step, we model the probability of expanding a new node from $v_t$ as:

$$p_t = \sigma(\text{MLP}(h_{v_t}, h_{\mathcal{G}_t}, \alpha_t^d)) \qquad \alpha_t^d = \text{attention}_d \left([h_{v_t}, h_{\mathcal{G}_t}], c_X^{\mathcal{G}}\right) \tag{13}$$

In the second step, the atom type of the new node $u_t$ is predicted using another MLP:

$$q_t = \text{softmax}(\text{MLP}(h_{v_t}, h_{\mathcal{G}_t}, \alpha_t^s)) \qquad \alpha_t^s = \text{attention}_s \left([h_{v_t}, h_{\mathcal{G}_t}], c_X^{\mathcal{G}}\right) \tag{14}$$

In the last step, we predict the bonds between $u_t$ and nodes in $\mathcal{Q} = a_1, \cdots, a_n$ sequentially starting with $a_1 = v_t$. Specifically, for each atom pair $(u_t, a_k)$, we predict their bond type (single, double, triple or none) as the following:

$$\begin{aligned} b_{u_t,a_k} &= \text{softmax}(\text{MLP}(h_{\mathcal{G}_t}, h_{u_t}^k, h_{a_k}, \alpha_t^b)) &\tag{15} \\ \alpha_t^b &= \text{attention}_b \left([h_{\mathcal{G}_t}, h_{u_t}^k, h_{a_k}], c_X^{\mathcal{G}}\right) &\tag{16} \end{aligned}$$

where $h_{a_k}$ is the atom representation of node $a_k$ and $h_{u_t}^k$ is the representation of node $u_t$ at the $k^{\text{th}}$ bond prediction. Let $N_k(u_t)$ be node $u_t$'s current neighbor predicted in the first $k$ steps. $h_{u_t}^k$ is computed as follows to reflect its local graph structure after $k^{\text{th}}$ bond prediction:

$$h_{u_t}^k = \text{MLP}\left(x_{u_t}, \sum_{w \in N_k(u_t)} \nu_{w,u_t}\right) \qquad \nu_{w,u_t} = \text{MLP}(h_w, x_{w,u_t}) \tag{17}$$

where $x_{u_t}$ is the atom feature of $u_t$ (i.e., predicted atom type) and $x_{w,u_t}$ is the bond feature between $w$ and $u_t$ (i.e., predicted bond type). Intuitively, this can be viewed as running one-step message passing at each bond prediction step (i.e., passing the message $\nu_{w,u_t}$ from $w$ to $u_t$).

AtomG2G is trained under the same variational objective as HierG2G, with the latent code $z$ sampled from the posterior $Q(z|X,Y) = \mathcal{N}(\mu_{X,Y}, \sigma_{X,Y})$ and $[\mu_{X,Y}, \sigma_{X,Y}] = \text{MLP}(\sum c_Y^{\mathcal{G}} - \sum c_X^{\mathcal{G}})$.

|  | logP ($\delta = 0.6$) | logP ($\delta = 0.4$) | QED | DRD2 |
|---|---|---|---|---|
| Training set size | 75K | 99K | 88K | 34K |
| Test set size | 800 | 800 | 800 | 1000 |
| Substructure vocabulary $|\mathcal{S}|$ | 478 | 462 | 307 | 307 |
| Average attachment vocabulary $|\mathcal{A}(\mathcal{S}_t)|$ | 3.68 | 3.50 | 3.62 | 3.30 |

Table 3: Training set size and substructure vocabulary size for each dataset.

## C EXPERIMENTAL DETAILS

**Data** The single-property optimization datasets are directly downloaded from the link provided in Jin et al. (2019). The training set and substructure vocabulary size for each dataset is listed in Table 3. We constructed the multi-property optimization by combining the training set of QED and DRD2 optimization task. The test set contains 780 compounds that are not drug-like and DRD2-inactive. The training and test set is attached as part of the supplementary material.

**Hyperparameters** For HierG2G, we set the hidden layer dimension to be 270 and the embedding layer dimension 200. We set the latent code dimension $|z| = 8$ and KL regularization weight $\lambda_{KL} = 0.3$. We run $T = 20$ iterations of message passing in each layer of the encoder. For AtomG2G, we set the hidden layer and embedding layer dimension to be 400 so that both models have roughly the same number of parameters. We also set $\lambda_{KL} = 0.3$ and number of message passing iterations to be $T = 20$. We train both models with Adam optimizer with default parameters.

For CG-VAE (Liu et al., 2018), we used their official implementation for our experiments. Specifically, for each dataset, we trained a CG-VAE to generate molecules and predict property from the latent space. This gives us three CG-VAE models for logP, QED and DRD2 optimization tasks, respectively. At test time, each compound $X$ is translated following the same procedure as in Jin et al. (2018). First, we embed $X$ into its latent representation $z$ and perform gradient ascent over $z$ to maximize the predicted property score. This gives us $z_1, \cdots, z_K$ vectors for $K$ gradient steps. Then we decode $K$ molecules from $z_1, \cdots, z_K$ and select the one with the best property improvement within similarity constraint. We found that it is necessary to keep the KL regularization weight low ($\lambda_{KL} = 0.005$) to achieve meaningful results. When $\lambda_{KL} = 1.0$, the above gradient ascent procedure always generate molecules very dissimilar to the input $X$.

**Ablation Study** Our ablation studies are illustrated in Figure 8. In our first experiment, we changed our decoder to the atom-based decoder of AtomG2G. As the encoder is still hierarchical, we modified the input of the decoder attention to include both atom and substructure vectors. We set the hidden layer and embedding layer dimension to be 300 to match the original model size.

Our next two experiments reduces the number of hierarchies in both our encoder and decoder MPN. In the two-layer model, molecules are represented by $c_X = c_X^{\mathcal{G}} \cup c_X^{\mathcal{A}}$. We make topological and substructure predictions based on hidden vector $h_{A_k}$ instead of $h_{S_k}$ because the substructure layer is removed. In the one-layer model, molecules are represented by $c_X = c_X^{\mathcal{G}}$ and we make topological and substructure predictions based on atom vectors $\sum_{v \in \mathcal{S}_k} h_v$. The hidden layer dimension is adjusted accordingly to match the original model size.

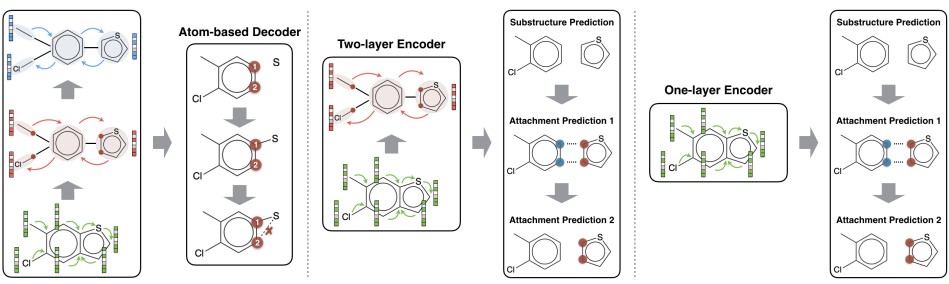

Figure 8: Model Ablations: 1) Atom-based decoder; 2) Two-layer encoder; 3) One-layer encoder.

