# OpenReview forum: "Hierarchical Graph-to-Graph Translation for Molecules"
_ICLR.cc/2020/Conference — Reject_

### Official Review · AnonReviewer1 · 2019-10-27
**Official Blind Review #1**

**Rating:** 3

**Review:**

This paper developed a hierarchical graph-to-graph translation model to generate molecular graphs using chemical substructures as building blocks. In contrast to previous work, the proposed model is fully autoregressive and learns coherent multi-resolution representations. The experimental results show that the proposed method outperforms previous models.

A few comments:

1.The novelty
- The method seems to be almost the same as the previous junction tree based formulation.  The paper includes a straightforward hierarchical extension and provides limited novelty with respect to deep learning.
- Can the method be used for other types of graph generation?

2. Some minor wording issues
- For instance, in the abstract, " In particular, we realize coherent multi-resolution representations .." What does this mean?

3. The main claim : " ... our graph decoder is fully autoregressive.."  why is this a merit?

4.  The paper provided results from multiple molecular optimization tasks. The results and analysis seem comprehensive. The model was shown to significantly outperform baseline methods in discovering molecules with desired properties. The model runs faster during decoding and can perform conditional translation.


**Experience Assessment:**

I have read many papers in this area.

**Review Assessment: Checking Correctness Of Derivations And Theory:**

I assessed the sensibility of the derivations and theory.

**Review Assessment: Checking Correctness Of Experiments:**

I assessed the sensibility of the experiments.

**Review Assessment: Thoroughness In Paper Reading:**

I read the paper at least twice and used my best judgement in assessing the paper.

---

> ### Author Response · Authors · 2019-11-10
> **Response (with a new figure in the paper)**
>
> Thank you for your insightful comments. We would like to first clarify the difference between our method and previous junction tree approach:
>
> Junction tree method of [Jin et al. 2019]:
>  - Two independently operating encoders, one for the junction tree, the other for the original graph
>  - Decoding is a strictly two-stage process (latent vectors -> junction tree -> graph) where the decoded junction tree is not affected by how the substructures are attached together
>  - The graph decoder operates locally and predicts attachments between connected substructures in the junction tree independently during training.
>
> The proposed approach:
>  - Our encoder is a unified hierarchical message passing network where lower levels directly impact higher level (substructure) representations.
>  - Previous attachment choices directly guide which substructures are/can be added to the expanding molecular structure.
>  - Our hierarchical decoder unravels substructure choices together with their attachments in an autoregressive manner.
>
>
> Q1: What’s the novelty / contribution of our approach?
> Our approach seeks to address two key limitations of the junction tree method [Jin et al. 2019], which is illustrated in Figure 1 in the paper (page 2):
>  - Their decoding is a strictly two-stage process. In the first stage, substructures are chosen without regard to how they can be attached to each other in the second stage. Therefore, it can result in invalid junction trees that cannot be realized into any molecule (see Figure 1a).
>  - Their local graph decoder can lead to inconsistent substructure attachments (see Figure 1b).
>
> The first problem is addressed by our hierarchical decoding process, where the predicted attachments are fed into decoder message passing network to guide substructure prediction. We solve the second problem by using an autoregressive decoder where previous attachment choices directly impact later substructure and attachment predictions.
>
> Q2: What’s the merit of autoregressive decoder?
> Autoregressive decoding prevents the model from predicting inconsistent substructure attachments (as illustrated by Figure 1b in the paper). As shown in the experiment, our autoregressive decoder indeed outperforms JTNN (e.g., 85.9% vs 77.8% on the DRD2 task).
>
> Q3: What does "coherent multi-resolution representations" mean?
> “Multi-resolution representation” refers to the substructure and atom embeddings. They together represent molecules at multiple levels. We used the word “coherent” because our substructure and atom encodings are learned by one hierarchical MPN instead of two separate MPNs as in the junction tree method.

---

### Official Review · AnonReviewer3 · 2019-10-31
**Official Blind Review #3**

**Rating:** 3

**Review:**

The authors present a heirarchical graph-to-graph translation method for generating novel organic molecules.
Working from the model of Jin et al. (2019), the authors introduce a three step heirarchy - the model first determines where a substructure should be generated, what is the substructure, then the attachments to the existing molecule.
All steps of this uses embeddings generated from a message passing network - these embeddings are input into a few bilinear attention layers to obtain the heirarchical generation scheme.
The model is trained with molecular pairs (X, Y), and a VAE loss - a hidden z vector controls the way to modify X to improve its properties.
The encoder is just a MLP over the difference between sum of embeddings at a atom level and at the substructure level.
The model is evaluated on accuracy and diversity, in both conditional and unconditional settings.
The experiments show a small improvement over previous SOTA algorithms.

This is a borderline paper, and I'm leaning towards a weak reject, because I don't believe the model is well motivated enough:
- Sec 3.1 it's unclear how the substructures are generated - they provide a lot of inductive bias for the algorithm.
  Are they automatically generated or built from a database of substructures?
- Variational decoding does not seem well motivated enough - would a stochastic decoding procedure not work as well as having a latent vector that essentially adds noise to the training?
- The experiments seem interesting and comprehensive - it seems that the model learns to exploit the biases and increase logP, as well as showing the ability to conditionally turn off DRD2-active properties of the molecules.

Some questions:
- Why not use a Transformer instead of an LSTM or GRU? The cell naturally acts over sets of neighbors and transformers are a natural model to tackle this problem.
- Sec 3.1 Topological Prediction, the attention is over c_{X}^{S} but the text claims it should be over c_{X}^{G}? Is ^G the attention substructure?
- Sec 3.1 Attachment Layer MPN: the A_i seem to be a tuple (S_i, {v_j}). The set of attaching atoms is limited to 2 right? It might be more clear to simply enumerate them here if so.
- Sec 3.1 Substructure Tree: Since tree decompositions are not unique, does this work use the different tree decompositions and DFS traversals as data augmentations?
- Table 2b: What is a "two-layer" and "one layer" encoder? Is it the size of the MLP or the removal of the attachment MPNs?
- Ablation study: Since the Attachment Layer has all the substructure information, this ablation should ideally make sure the models all have a similar number of parameters, and the decrease in performance isn't due to the decrease in parameters.

Nits:
- Sec 3.1 "bi-linear" should not have a dash, bilinear is one word.


**Experience Assessment:**

I have read many papers in this area.

**Review Assessment: Checking Correctness Of Derivations And Theory:**

N/A

**Review Assessment: Checking Correctness Of Experiments:**

I assessed the sensibility of the experiments.

**Review Assessment: Thoroughness In Paper Reading:**

I made a quick assessment of this paper.

---

> ### Author Response · Authors · 2019-11-10
> **Response (with a new figure in the paper)**
>
> Thank you for your insightful comments. We want to first explain the motivation of our approach. The proposed hierarchical architecture seeks to address two key limitations of the junction tree method [Jin et al. 2019], which is illustrated in Figure 1 in the paper (page 2):
>  - Their decoding is a strictly two-stage process (latent vectors -> junction tree -> graph). In the first stage, substructures are chosen without regard to how they can be attached to each other in the second stage. Therefore, it can result in invalid junction trees that cannot be realized into any molecule (see Figure 1a).
>  - Their local graph decoder can lead to inconsistent substructure attachments (see Figure 1b).
>
> The first problem is addressed by our hierarchical decoding process, where the predicted attachments are fed into decoder message passing network to guide substructure prediction. We solve the second problem by using an autoregressive decoder where previous attachment choices directly impact later substructure and attachment predictions.
>
> Some clarifications:
>  - The encoder is the hierarchical message passing network that outputs substructure and atom vectors (not MLP). MLP is the variational inference module that learns the latent vector z, which is in addition to the message passing network.
>  - The model achieves pretty large improvements on some tasks. For example, on the DRD2 task our model shows large improvement over previous SOTA (77.8% -> 85.9%).
>
> Q1: How are the substructures generated?
> Substructures are automatically extracted from the molecules in the training set, in order to ensure structural coverage. For a given molecule, we extract its (smallest) rings and bonds as substructures and add them to the vocabulary. This procedure is purely data-driven.
> Indeed, the substructure vocabulary provides a lot of inductive bias for the algorithm and optimizing the vocabulary for downstream task is an interesting future work.
>
> Q2: Variational decoding is not well motivated. Why not using a stochastic decoding procedure?
> We used variational decoding for two reasons:
>  - All the prior work (Seq2Seq and JTNN) used variational decoding. Therefore we adopted the same strategy to establish a direct comparison.
>  - Recent work has shown that variational decoding can generate more diverse outputs than stochastic decoding (e.g., in image translation [Zhu et al., 2017] and machine translation [Shen et al., 2019]). The reason is that stochastic decoding tends to learn small, local variations (e.g., replacing single atoms), while variational decoding captures diversity beyond local variations. To show this, we trained our model without variational inference and used stochastic decoding at test time. On the logP (sim $\geq$ 0.6) dataset, the performance drops from 2.49 to 2.06 and diversity drops from 0.381 to 0.342. On the logP (sim $\geq$ 0.4) dataset, the performance drops from 3.98 to 3.72 and diversity drops from 0.564 to 0.502.
>
> Q3: Why not use a Transformer instead of an LSTM or GRU?
> We used LSTM / GRU because message passing networks (MPN) are standard choices for molecules and many previous works build upon MPNs (with various parameterizations). We agree that transformer is a promising architecture for graphs (especially if further tailored to graphs) but the gains are unclear at this point.
>
> Q4: Topological Prediction: the attention is over $c_{X}^{S}$ but the text claims it should be over $c_{X}^{G}$?
> We apologize for the confusion. This is a typo and it should be $c_{X}^{S}$. The typo is now fixed.
>
> Q5: Attachment Layer MPN: the A_i seem to be a tuple $(S_i, {v_j})$. The set of attaching atoms is limited to 2 right?
> The set of attaching atoms can be more than 2 because two rings can have three or more overlapping atoms.
>
> Q6: Since tree decompositions are not unique, does this work use different tree decompositions and DFS traversals as data augmentations?
> We didn’t use different tree decompositions / DFS traversals for data augmentation because none of the baselines used any data augmentation strategies.
>
> Q7: Table 2b: What is a "two-layer" and "one-layer" encoder?
> Two-layer encoder means the top substructure layer MPN is removed. One-layer encoder means the attachment layer MPN is also removed.
>
> Q8: Ablation studies, number of parameters.
> For ablation studies, all models have a similar number of parameters. For both datasets, all the model parameters are between 6M to 6.2M.
>
> References
> Zhu et al. "Toward multimodal image-to-image translation." Advances in Neural Information Processing Systems. 2017.
> Shen et al. "Mixture Models for Diverse Machine Translation: Tricks of the Trade." International Conference on Machine Learning. 2019.

---

### Official Review · AnonReviewer4 · 2019-11-01
**Official Blind Review #4**

**Rating:** 6

**Review:**

This paper proposed a hierarchical graph-to-graph translation method to modify compounds to improve the biochemical properties. The authors proposed to generate the new molecular in a autoregressive manner. To improve the performance of the model, the input molecular is encoded into different resolutions including atom, attachment, and substructure layer. The paper is well written and the figures in the paper also enable the paper easy to read. In the experiments, the authors compare the proposed method with serval state-of-the-art methods. The results well analyzed and the ablation study is provided. Overall, this is a good paper considering its technical contribution and writing.

However, there are some small issues should be addressed:

1. There are some typos in the paper. For example, in the topological prediction section in page 3,  "the hidden representation of $S_k$ learned be the decoder " -> learned be encoder.

2. In the training, the authors apply teacher forcing to the generation process with depth-first order. Why do you use depth-first order not any other orders?


**Experience Assessment:**

I do not know much about this area.

**Review Assessment: Checking Correctness Of Derivations And Theory:**

I assessed the sensibility of the derivations and theory.

**Review Assessment: Checking Correctness Of Experiments:**

I carefully checked the experiments.

**Review Assessment: Thoroughness In Paper Reading:**

I read the paper thoroughly.

---

> ### Author Response · Authors · 2019-11-10
> **Response**
>
> Thank you for your insightful comments. We have fixed the typo you pointed out (and some others).
>
> Q1: In the training, the authors apply teacher forcing to the generation process with depth-first order. Why do you use depth-first order instead of other orders?
> Our hierarchical decoding process leverages the fact that graphs are generated in a depth-first order, especially topological prediction step. To use other orders such as breadth-first order, the graph decoding process needs to be modified accordingly. We chose depth-first order in order to make a fair comparison with JTNN, which also uses depth-first order.

---

### Decision · Program_Chairs · 2019-12-19

**Decision:**

Reject

**Comment:**

Two reviewers are negative on this paper while the other reviewer is slightly positive. Overall, the paper does not make the bar of ICLR. A reject is recommended.